# Parental Stress in Raising a Child with Developmental Disabilities in a Rural Community in South Africa

**DOI:** 10.3390/ijerph20053969

**Published:** 2023-02-23

**Authors:** Nontokozo Lilian Mbatha, Kebogile Elizabeth Mokwena

**Affiliations:** 1Department of Public Health, Sefako Makgatho Health Sciences University, Pretoria 0001, South Africa; 2NRF Chair in Substance Abuse and Population Mental Health, Department of Public Health, Sefako Makgatho Health Sciences University, Pretoria 0001, South Africa

**Keywords:** parental stress, developmental disabilities, caregivers, parenting stress index short form

## Abstract

Although acceptable levels of parental stress are experienced by all parents who raise children, this stress is substantially higher among parents who raise children with developmental disabilities. Sociodemographic determinants further exacerbate parental stress among parents in rural communities, which are disadvantaged in many ways. This study aimed to quantify parental stress among mothers and female caregivers of children with developmental disorders and investigate factors associated with such stress in rural Kwa-Zulu Natal, South Africa. A cross-sectional quantitative survey was used, in which the Parenting Stress Index-Short Form (PSI-SF) and a sociodemographic questionnaire was administered to mothers and caregivers who were raising children aged 1 to 12 years old who were living with developmental disabilities. The PSI-SF scores were used, where a total score of ≤84 percentile was categorised as normal/no parenting stress, 85–89 percentile was categorised as high parental stress, and scores of ≥90 were classified as clinically significant. The sample of 335 participants consisted of 270 (80.6%) mothers and 65 (19.4%) caregivers. Their ages ranged from 19 to 65 years, with a mean of 33.9 (±7.8) years. The children were mostly diagnosed with delayed developmental milestones, communication difficulties, epilepsy, cerebral palsy, autism, ADHD, cognitive impairment, sensory impairments, and learning difficulties. The majority (52.2%) of the participants reported very high-clinically significant stress levels (≥85%ile). The four factors that independently and significantly predicted high parental stress were the advanced age of mothers and caregivers (*p* = 0.002, OR 2.3, 95% CI 1.34–3.95), caring for a child with multiple diagnoses (*p* = 0.013, OR 2.0, 95% CI 1.16–3.50), non-school enrolment of the child (*p* = 0.017, OR 1.9, 95% CI 1.13–3.46), and frequent hospital visits (*p* = 0.025, OR 1.9, 95% CI 1.09–3.44). At the subscale level, child non-enrolment in a school was found to independently predict parent distress (PD) and parent-child dysfunctional interaction (P-CDI). Frequent hospital visits were statistically and significantly associated with the difficult child (DC) and P-CDI subscales. The study established high parental stress in mothers and caregivers raising children with developmental disabilities. Lack of access to school was an independent factor that consistently increased parental stress. There is a need for support and directed intervention programs aimed at supporting mothers and caregivers of children with developmental disabilities, which will enhance their parenting abilities.

## 1. Introduction and Background

The burden of developmental disability due to impairment in physical, learning, language or behaviour is increasing globally, with many children living with disabling and long-term health conditions. The affected children experience compromised optimal development and lack adequate access to essential health and education opportunities that can serve their specific needs [1,2,3,4]. Most children with disabilities live in low-income and middle-income countries (LMICs), where they are marginalised and vulnerable to neglect and live in poverty, abuse, and violence [5].

Being a parent of or raising a child comes with normal parental stress, which is distinct and emanates explicitly from a range of responsibilities or roles associated with being a parent [6]. While parental stress applies to all parents, parenting stress for parents and caregivers raising children with developmental disabilities has been reported to be incredibly challenging; its impacts are further exacerbated by socioeconomic status. Families of low socioeconomic status often lack information and understanding about their children’s condition and experience uncertainty and disappointment about their children’s future and developmental restrictions [7]. Moreover, parents may feel overwhelmed and burdened and even find it challenging to follow treatment and rehabilitation plans recommended for their children [6,8], subsequent increased parental stress.

Parenting and childhood are intrinsically related, occurring within a socially constructed environment which is predominantly bound by cultural and socioeconomic status. This then raises the crucial need to understand the relationship between parenting and the parenting context. A child’s developmental context, as determined by the physical and social environment, is the strongest predictor of the child’s developmental outcomes. Meaning that a compromised home environment negatively influences the child’s cognitive and social development. Therefore, the analysis of a child’s development should extend to the family and the social context in the child’s life [9]. The social environment was found to be essential for the exploration of parental competence and self-efficacy and is significant in influencing parenting outcomes [10,11].

The family is a critical component of a child’s socialisation, thus a need to conceptualise and understand disability within the geographic and social context, including the African context. In many African communities, the family bonds are not only between the biological parents and the child. However, they are a very complex phenomenon that includes extended family members and siblings [12], which can present both positive and negative attributes. Most African families lack institutional services and support in raising their children with disabilities. Unfavourable parenting environments frustrate the ability of the family to foster the development of their children with developmental disabilities [13].

Although there are many contributory factors and histories behind child developmental disabilities, they often co-occur with chronic health conditions. On the other hand, several chronic health conditions contribute to disability [14]. Regardless of contributing factors, developmental disability and chronic health conditions often have a negative emotional burden on children and their families, including costs for health and social services [15,16]. Thus, parents who have children with developmental disabilities or chronic health conditions have high levels of parenting stress than parents of children without disabilities [8,17,18,19].

The emotional and support needs of children with disabilities and their families are dynamic and often complex. Parents raising a child with a disability often face numerous challenges relating to social isolation, emotional stress and depression, grief and financial problems. Recent studies suggest that some cultural beliefs exacerbate stigmatisation [20,21], which further increases parental stress. Setbacks experienced by families who are raising children with disabilities include, among other financial constraints, lack of support, lack of information, and transport barriers that intensify the burden upon the parents and caregivers and affect how they meet the needs of their children [22].

A range of social environment factors, including sociodemographic status, maternal education, parental self-efficacy, and access to knowledge on the child’s development, directly influence parental competence to promote the child’s physical and social well-being and thus determine the child’s developmental outcomes. Other sociodemographic variables related to child development include family income and parental educational level [23]. A poor economic situation may limit the child’s access to resources they may need for their development [24], resulting in poor outcomes. On the other hand, parents with higher levels of education have been consistently associated with seeking, and affording high-quality childcare programs, thus fostering better developmental outcomes for their children [25,26].

Although reports of uniquely high parental stress on parents and caregivers raising children with disabilities, there is a dearth of studies specific to the South African context. Moreover, South African rural communities are reported to have increased levels of poverty and inadequate education [27,28], as well as inadequate health and social services [29], all of which increase parental stress. These factors imply even more challenges for the parents’ adequacy to raise children with disabilities. This study aimed to quantify parental/caregiver stress among parents raising children with disabilities in rural communities in Kwa-Zulu Natal, South Africa. The article was drawn from a more extensive study that screened for parental stress, perceived self-efficacy of parents or caregivers concerning their ability to parent a child with developmental disabilities and determine the family structure upon which parenting occurs.

## 2. Study Methodology

### 2.1. Study Design

A cross-sectional quantitative survey was used, in which a sociodemographic questionnaire and Parenting Stress Index-Short Form were administered to mothers and caregivers who were raising children with developmental disabilities.

### 2.2. Study Setting

The study was conducted in a regional public hospital that is dedicated to women and children under the age of 12 years receiving services in King Cetshwayo District (KCD) of Kwa Zulu Natal province, South Africa. Although the rural KCD is characterised by poverty and unemployment, they have access to primary health care through the residents’ primary health care clinics or mobile clinics. Most of the KCD population depends on government public health services, and only 8.7% have access to medical aid. The district has one public tertiary institution, one regional hospital, and the mother and child hospital where the study was conducted. The regional hospital is a referral facility for 16 district hospitals within the region. The specialised neurodevelopmental clinic in this hospital provides services to more than 1500 children annually for follow-ups or initial medical assessment and treatment.

### 2.3. Population of the Study

The study population consisted of mothers and caregivers of the estimated 1500 children aged 1–12 years who have developmental disabilities and attend the specialised neurodevelopmental clinic and rehabilitation services in this regional hospital. A caregiver should have lived/cared for the child in the past six months.

### 2.4. Sample Size

Using the Raosoft sample size calculator for an estimated population size of 1500, a confidence level of 95% and a 5% margin of error, a minimum sample size of 306 was calculated. The final total sample size of mothers and caregivers was three hundred and thirty-five (*n =* 335).

### 2.5. Recruitment

Upon obtaining ethical clearance and permission from the Sefako Makgatho Health Science University Research Ethics Committee (SMUREC) and the Province of KwaZulu Natal Health Research and Knowledge Management Committee, respectively, and then the regional hospital for permission to collect data. Within the neuro-developmental clinic, the researcher approached mothers and caregivers who were visiting the clinic for services required for their children while they were waiting in the outpatient department and recruited them to participate in the study. Those who agreed were directed to a separate room where data collection took place.

### 2.6. Inclusion and Exclusion Criteria

Participants in this study were IsiZulu-speaking mothers raising a child with developmental disabilities, caregiver needed to have been with the child for at least six months and have a full understanding of the child’s health needs. In this study, inclusion criteria also related to the demographic characteristics of the child (the children were aged between 1–12 years, male or female gender). Clinical characteristics (reported to have a diagnosis of developmental disorder, impairments, and currently accessing neurodevelopmental clinic at the hospital where data was being collected). Exclusion criteria related to acquired medical conditions such as surgical injuries and other childhood diseases.

### 2.7. Data Method and Tools

The data was collected by the researcher and a trained researcher assistant. On the day of data collection, the individual participants who were invited and agreed to participate in the study were invited to a private space where their privacy was ensured. The purpose of the study was explained, and the participant was given an opportunity to ask questions or seek clarifications, which was followed by the administration of the informed consent. A researcher-developed socio-demographic questionnaire was administered first, which was followed by the PSI-SF. Participants who could administer the tools were encouraged to do so, and those who could not be assisted by the researcher or research assistant. The tools were in isiZulu, which is the common language used in the region. The data was collected over a period of eight months.

### 2.8. The Parental Stress Index—Short Form (PSI-SF)

The PSI-SF is a valid and reliable measurement tool derived from the longer Parental Stress Index long-form of 120 items [30]. It is a 36-item representing three domains: parental distress (PD), parent-child dysfunctional interaction (P-CDI) and difficult child (DC) subscales. Total parenting scores based on all three domains range from 36–180. Parental stress percentile scores within the 16–84th percentile are considered normal. Scores in the 85–89th percentile are considered high, and scores ≥ 90th percentile are considered to be clinically significant. PSI-SF is a valid and reliable tool to measure parental stress; it has been used in South Africa [31] and in Africa [32,33] in a rural community. A standardised isiZulu version of the PSI-SF was used in this study.

### 2.9. Adherence to COVID-19 Guidelines

The researcher followed the COVID-19 regulations and adhered to the institutional guidelines. Such guidelines included the wearing of protective clothing, frequent washing of hands, and the use the sanitiser for all participants and the researcher/assistant.

### 2.10. Data Analysis

STATA 17 was used to analyse data. Covariates were presented as mean values (standard deviations). Sociodemographic data were analysed descriptively and presented as frequencies, proportions and percentages. The PSI-SF percentile scores determined the presence and severity of parental stress. The total score on the PSI-SF was first analysed according to the three categories as outlined in the manual normal (16–84th percentile), high (85–89th percentile), and clinically significant (≥90th percentile). The total score on the PSI-SF was further dichotomised into the ≤84% percentile and ≥85% percentile, the latter indicating high-clinically significant stress. Pearson’s chi-square association test explored associations between parental stress levels and sociodemographic variables. Multivariate logistic regression analysis was conducted for the sociodemographic variables statistically significantly associated with parental stress measures in the three subscales and the total stress, using a *p*-value of 0.05.

### 2.11. Ethical Considerations

This study received ethical clearance from the Sefako Makgatho Health Science University Research Ethics Committee (SMUREC) and the Province of KwaZulu Natal Health Research and Knowledge Management Committee. Permission to conduct the study was obtained from the hospital’s ethical committee, and informed consent was received from mothers and caregivers.

## 3. Results

### 3.1. Socio-Demographic of the Mothers and Caregivers

The 335 participants consisted of 270 (80.6%) mothers and 65 (19.4%) caregivers. Their ages ranged from 19 to 65 years, with a mean of 33.9 (±7.8) years. The majority (59.7%, n = 200) were younger than 35 years, single (86%, n = 288) and resided in a rural community (89.2%, n = 299). Sixty-five (19.4%) were caregivers/not biological parents; most caregivers were grandparents. The rest of the socio-demographic variables are reflected in Table 1 below.

### 3.2. Sociodemographic Characteristics of the Children

The children cared for were aged 1 to 12 years, with a mean age of 5.6 (±3) years. Most of the children were boys (69.1%, n = 231), 4 years and younger (42.1%, n = 141), born in a hospital (82.5%, n = 285), with a history of previous hospitalisation (59.4%, n = 198), and residing in King Cetshwayo district (70.6%, n = 236). The children were reported by the mothers and caregivers as diagnosed with developmental delayed milestones (41.4%, n = 138), communication difficulties (31.8%, n = 106), epilepsy (23.2%, n = 78), cerebral palsy (18.7%, n = 63), and autism (9.6%, n = 31). Other diagnoses reported by mothers and caregivers included ADHD, mental, sensory impairments, and learning difficulties.

Most participants (73%, n = 245) reported visiting the hospital monthly for healthcare services, including rehabilitation services (62.2%, n = 208), and 58.8% (n = 196) visited the tertiary hospital for the specialised level of care. Access to school, either at a special school level or a mainstream school, was reported for only 35.3% (n = 118) of the children. The majority (64.3%, n = 217) reported not attending for various reasons, including the unavailability of suitable schools and financial problems. Among those not attending school, 35% (n = 76) were ≥5 years old and awaiting school placement. Children enrolled at school were reported to be at local mainstream or at creche (69.1%, n = 75) and only 30.1% (n = 33) at a school for children with special needs. The rest of the socio-demographic variables are reflected in Table 1 below.

### 3.3. Prevalence of Parental Stress

The parental stress index scores ranged from 48 to 178, with a mean of 116 (±23.5) scores. Results showed that 52.2% (n = 175) of mothers’ and caregivers’ scores were clinically significantly stressed, 6.3% (n = 21) were highly stressed, and 41.5% (n = 139) were not stressed.

Table 2 below outlines the different subscales scores of the PSI-SF. Parental distress had a significantly high mean of 40.7 (±10.3) compared to the other subscales. Most clinically significant scores were observed in the DC subscale (55.5%, n = 221) and PD (54%, n = 181).

### 3.4. Parental Stress Levels and Sociodemographic Characteristics of Mother and Caregiver

The PSI scores were first determined at ≤84 percentiles (normal), 85–89 percentile (high parental stress) and ≥90 percentiles (clinically significant scores) and were then dichotomised into normal (≤84 percentiles) and high-clinically significant (≥85 percentiles). As shown in Table 3 below, high-clinical significant stress was reported by 58.5% (n = 196) of the mothers and caregivers. The majority were age 36 years and above (65.9%, n = 89), married (59.6%), in a rural area (59.9%), non-Christian (60.4%), with no matric (60.2%), employed (62%), earning ≥ R5000 (65%), and caregivers who were not financially supported by their partners (74.2%).

Table 3 shows the socio-demographic variables of the children, and high to clinically significant levels of stress scores were observed in mothers and caregivers whose children were aged between 5–8 years (60.8%), boys (62.8%), born at home (83.3%), with no history of hospitalisation at birth (65%), had multiple (more than one) diagnosis of developmental disorders (66%), mostly with communication difficulties and delayed developmental milestones (71.7% & 67.4%) respectively, and visiting the hospital for health services monthly (63.7%).

### 3.5. Factors Associated with Parental Stress

#### 3.5.1. Subscale Levels

Table 4 shows the association between the sociodemographic factors to parental stress at subscale and total parental stress scores. Parental distress (PD) was significantly associated with the age of the mother and caregivers (*p* = 0.00), multiple health needs (*p* = 0.00) and school enrolment (*p* = 0.03). Several factors were found to be associated with DC. Mother and caregiver-related factors were the places of residence (*p* = 0.05) and level of education (*p* = 0.04). Child-related factors were the child’s gender (*p* = 0.03), history of hospitalisation (*p* = 0.02), having more than one (multiple) diagnoses (*p* = 0.00), communication difficulties (*p* = 0.00), other disorders reported (*p* = 0.03), and the frequency of hospital visits (*p* = 0.03) were found to be significantly associated with the mothers and caregiver’s perception of the child’s difficult behaviour (DC). Similarly, P-CDI had common factors with DC, and this included the history of hospitalisation after birth (*p* = 0.02), diagnosis of multiple disorders (*p* = 0.01), communication difficulties (*p* = 0.01), diagnosis with other disorders (*p* = 0.05), and frequency of hospital visits (*p* = 0.00). Unlike other subscales, P-CDI was found to be significantly associated with the age of initial intervention (*p* = 0.03) and diagnosis of delayed developmental milestones (*p* = 0.01).

#### 3.5.2. Total Parental Stress Levels

Child-related factors found to predict high total parental stress scores were the child’s gender (*p* = 0.01), history of hospitalisation (*p* = 0.04), diagnosis with more than one developmental disorder (*p* = 0.02), communication difficulties (*p* = 0.00), diagnosis of delayed developmental milestones (*p* = 0.00), having multiple health needs (*p* = 0.01), frequency of hospital visit (*p* = 0.00), and the child school enrolment (*p* = 0.04). Differently from the subscale’s factors, visiting tertiary-level hospitals for specialised care was found to be associated with high total parental stress (*p* = 0.02). The age of mothers and caregivers (OR 1.8, 95% CI 1.09–2.79, *p* = 0.024) was found to be statistically significantly associated with parental stress.

The following child-related factors were statistically significantly associated with high total parental stress, the child’s gender (OR 1.8, 95% CI 1.09–2.79, *p* = 0.018), diagnosis with more than one disorder (OR 2.3, 95% CI 1.44–3.55, *p* = 0.001), communication difficulties (OR 2.3, 95% CI 1.40–3.77, *p* = 0.001), delayed developmental milestones (OR 1.9, 95% CI 1.19–2.93, *p* = 0.013), the frequency of hospital visit (OR 2.2, 95% CI 1.34–3.58, *p* = 0.002) and school enrolment (OR 1.2, 95% CI 0.98–2.42, *p* = 0.040).

Although the association was significant for the following factors, it was however not statistically significantly associated with parental stress. The history of hospitalisation (OR 0.6, 95% CI 0.40–0.99, *p* = 0.046), child’s multiple health needs (OR 0.6, 95% CI 0.35–0.89, *p* = 0.001) and visiting tertiary level hospitals visits (OR 0.8, 95% CI 0.53–1.29, *p* = 0.002). These factors were less likely to increase parental stress, as the odds ratio was below one.

### 3.6. Multivariate Logistic Regression

Variables that were statistically significantly associated with parental stress at subscales and total parental stress were included in the multivariate logistic model. The analysis began with the analysis at the subscale level and, lastly, the analysis of factors that predicts total parental stress.

#### 3.6.1. Multivariate Logistic Regression: Subscale

At the Chi-square test of association level, parental distress was statistically significantly associated with the age of mothers and caregivers, multiple health needs and school attendance. These variables were used to build a multivariate logistic regression model for the PD subscale. Two factors independently predicted parental distress. The child school enrolment (*p* = 0.003, OR 2.1, 95% CI 1.31; 3.64) and the age of mothers and caregivers (*p* = 0.001, OR 2.5, 95% CI 1.49; 4.22) were statistically significantly associated with parental distress.

Difficult child at the chi-square level was statistically significantly associated with place of residence, level of education, gender of the child, history of hospitalisation, multiple diagnoses, communication difficulties, diagnosis with other disorders, and frequency of hospital visits. These factors were used to build a multivariate logistic regression model for the DC subscale. Three factors were found to independently predict the DC subscale. Communication difficulty (*p* = 0.042, OR 1.9, 95% CI 1.02–3.58), the child’s diagnoses with more than one developmental disorder (*p* = 0.008, OR 2.1, 95% CI 1.21–3.67) and the frequency of hospital visits (*p* = 0.045, OR 1.8, 95% CI 1.01–3.16).

Seven variables were purposefully included in this multivariable regression model to determine factors that predicted P-CDI. These were variables found to be significant at the Chi-square analysis level, history of hospitalisation, multiple diagnoses, communication difficulty, developmental disorders, other diagnoses, age of intervention, school attendance and sanitation facility. In this current study, P-CDI was found to be statistically significantly predicted by two factors. The frequency of hospital visits (*p* = 0.018, OR 1.9, 95% CI 1.12; 3.25) and the child’s school enrolment (*p* = 0.002, OR 2.4, 95% CI 1.37; 4.09).

Overall at the subscale level (Table 5), child non-enrolment to a school was found to independently predict parent distress and parent-child dysfunctional interaction. The frequency of hospital visits (monthly hospital visits) statistically and significantly predicted DC and P-CDI. Different from the other subscale, the age of the mother and caregiver was found to predict PD independently. A diagnosis of delayed developmental milestones and communication difficulties is statistically and significantly associated with DC.

The final multivariate logistic regression model included nine factors that were statistically significantly associated with the total parental stress scores (gender of child, multiple diagnoses, developmental delay, and multiple health needs) and independently predicted the subscale scores (age of mother and caregiver, school, multiple diagnoses, communication difficulties, and frequency of hospital visit). Four factors independently predicted high parental stress. The age of mothers and caregivers ≥36 years (*p* = 0.002, OR 2.3, 95% CI 1.34; 3.95), caring for a child with multiple diagnoses (*p* = 0.013, OR 2.0, 95% CI 1.16; 3.50), non-school enrolment (*p* = 0.017, OR 1.9, 95% CI 1.13; 3.46), and monthly hospital visit (*p* = 0.025, OR 1.9, 95% CI 1.09; 3.44) were statistically and significantly associated with increased total parental stress (Table 5).

#### 3.6.2. Discussion

This study describes stress among mothers and caregivers of children with developmental disorders and explores factors associated with high parenting stress. Parenting stress levels in mothers and caregivers raising children with developmental disabilities were significantly high. The results showed that 52.2% of mothers and caregivers reported elevated total PSI/SF scores. These high scores indicate that mothers and caregivers raising children with developmental disabilities are at risk of increased parenting stress. The results are consistent with recent studies in South Africa and the Sub-Saharan region. In a recent study done in Gauteng, South Africa, the majority (87.5%) of the primary caregivers of children with cerebral palsy suffered from clinically significant stress [34]. Recent studies conducted in Uganda and Kenya reported that over half of parents raising children with developmental disabilities experience high parenting stress [35,36]. In a study with Taiwanese mothers, Ref. [37] found that mothers raising a child with autism experience high parental stress compared to parents raising children with developmental delay, which suggests that different types of developmental disability influence parental stress levels.

In this study, factors that relate to the parent’s view of the child’s temperament, defiance, noncompliance, and demandingness, were the most provoking factors increasing parental stress in mothers and caregivers raising children with developmental disabilities, which supports previous findings that reported these as provoking factors that increased parenting stress [30]. In the current study, the age of mothers and caregivers caring for a child with multiple disorders, frequency of hospital visits and non-school enrolment independently predicted parental stress.

The relationship between chronic health conditions and developmental disabilities in children is complex, as several chronic health conditions contribute to disability, making the child’s condition complex [14]. The current study found that mothers and caregivers caring for children with more than one developmental disorder are at increased risk for parental stress. Similarly, other studies found that high parenting stress was associated with more caregiving demands in children with severe chronic conditions, gross motor difficulties, behavioural problems (ADHD) and communication difficulties [38,39,40,41].

Literature has reported that multiple diagnoses have an increased emotional burden on the parents, as well as a financial burden for health-associated services [16,42]. The study found that frequent hospital visits for health-related needs associated with the child’s diagnosis doubled the risk of parenting stress among the participants. Mothers and caregivers who were visiting the hospital monthly were at a higher risk for parenting stress. There is a financial burden linked to monthly transport costs. However, for parents of children with developmental disabilities, monthly hospital visit means being in a public setting more frequently and using public transport with the child. Using public transport is known to be challenging for children with developmental disabilities [43]. Moreover, children with disabilities are prone to hospitalisation, during which parental stress is heightened [44]. Little is known about the effect of frequent outpatient visits on parents of children with disability. To the best of our knowledge, this study is the first to establish with evidence the association between parental stress and frequent outpatient visits in South Africa.

Education is essential, and lack of access to it compromises optimal development [3,4]. The study found a lack of schooling as a risk factor for parental stress. Section 29 of the South African Constitution identifies access to education as a right of every citizen and identifies the State as the responsible entity to ensure this right [45]. Failing to provide schooling for children with disabilities is, therefore, a direct violation of the constitution by the Department of Basic Education. The unavailability of suitable schools, which is a reality in many rural communities of South Africa [46], has been documented, even for mainstream education. In South Africa, children with disabilities are five times more likely to be out of school when compared to the general population [47]. In this study, children with disabilities that were either too young to enrol at school and school-aged children that were not enrolled in any school due to financial problems or unavailability of suitable schools independently increased the likelihood of parenting stress. Mothers and caregivers caring for children with developmental disorders full-time were almost two times more likely to be stressed than those able to send their children to school. Barriers to education for children with developmental disorders are the greatest challenge and a significant stressor and are thus an area that needs interventions.

Parenting stress outcome was independently associated with the age of the mother or caregiver, where older mothers and caregivers reported higher stress compared to those who were 35 years and younger. The finding that the age of mothers and caregivers are statistically associated with parental stress is similar to that found among parents of children with spinal bifida [35], which suggests that any deviance from the normal development of a child increases parental stress of the parents. Older mothers and caregivers were found to have deeper concerns and resultant higher stress levels. These results are significant for this study as many of the caregivers were grandparents of the child. Over and above parental stress related to the developmental challenges of their children, some mothers were raising their children without the support or involvement of their partners, which increased stress levels.

### 3.7. Strengths and Limitations

One of the strengths of this study was the larger sample size from which the quantitative results were drawn. This study used 335 participants who were mothers and caregivers of children with developmental disorders. This study sample, however, did not include the fathers of the children. This study only focused on the parental stress of mothers and caregivers raising children with disabilities; hence a comparison with those raising typically growing children could not be made. The study population was dominantly African, isiZulu-speaking women. Therefore, the results cannot be generalised to other races. The child’s diagnosis was as reported by the mother or caregiver, and no patient information reviews were done. Future research studies should also consider the severity of the child’s disability.

### 3.8. Recommendations

It is recommended that support for mothers and caregivers who are raising children with disabilities be integrated into the services intended to benefit the child. The parenting capacity of mothers with elevated stress levels is compromised, with further disadvantages for the development of the child. Interventions to address the stress levels of the mother are, therefore, a necessity to support the affected child.

## 4. Conclusions

The study established high parental stress in mothers and caregivers raising children with developmental disabilities. Because stress compromises function and social interactions, the affected mothers’ ability to provide the necessary support for their children is decreased and therefore needs attention. In particular, not only does lack of access to school an independent factor that consistently increases parental stress, but it also violates the rights of the children, who should, as per the requirements of the constitution, be provided with quality education for their needs. Mental health services should therefore include the provision of services that target mothers who are raising children with disabilities. Specific interventions are also required for custom-made services for the children and their parents.

## Figures and Tables

**Table 1 ijerph-20-03969-t001:** Demographic characteristics of the participants.

**Mother and Caregiver Characteristics**		**n (335)**	
		** *n* **	** *%* **
Age of Mother & caregiver: mean (33.9), SD (7.8)	≤35 years	200	59.7%
	≥36 year	135	40.3%
Marital status	Married	47	14%
	Single	288	86%
Place of resident	Urban	36	10.8%
	Rural	299	89.2%
Level of education	Did not complete matric	108	32.2%
	Completed matric	166	49.6%
	Completed tertiary	61	18.2%
Employment status	Yes	50	14.9%
	No	285	85.1%
Estimated family monthly income	≥R5001	40	11.9%
	R2001-R5000	10	3%
	Care dependency grant	285	85.1%
Income source	Employment	50	14.9%
	Care Dependency Grant	285	85.1%
Supported by father of the child	Yes	304	90.8%
	No	31	9.2%
Child Related Characteristics			
Age of child	≤4 years	141	42.1%
	5–8 years	120	35.8%
	≥9 years	74	22.1%
Gender	Girls	104	30.9%
	Boys	231	69.1%
From the local district (KCD)	Yes	236	70.6%
	No	99	29.4%
Place of birth	Hospital	285	85.2%
	Clinic	38	11.3%
	Home	12	3.5%
History of hospitalisation	Yes	198	59.4%
	No	137	40.6%
Multiple diagnoses	Yes	200	60.8%
	No	130	39.2%
Developmental disability	Cerebral palsy	63	18.7%
	Autism	31	9.6%
	Communication	106	31.8%
	Delayed developmentalmilestones	138	41.4%
	Epilepsy	78	23.2%
	Other diagnoses	108	32.4%
Main health needs	Rehabilitation	208	62.2%
	Multiple Needs	215	64.6%
Age of initial intervention	≤2 years	247	73.3%
	≥3 years	88	26.7%
Frequency of hospital visits	Monthly	245	73%
	Once (2/6 months)	90	27%
Tertiary level hospital visits	Yes	196	58.8%
	No	139	41.2%
Attending school	Yes	118	35.3%
	No	217	64.7%

**Table 2 ijerph-20-03969-t002:** Parental stress subscales.

	Mean (SD)	Normal (≤84%ile)	High (85–89%ile)	Clinically Significant (≥90%ile)
Parental Distress	40.7 (10.3)	136 (40.6%)	18 (5.4%)	181 (54%)
Parent-Child Dysfunctional Interaction	37.4 (9.5)	125 (37.3%)	48 (14.3%)	162 (48.4%)
Difficult Child	37.9 (8.4)	114 (34%)	35 (10.5%)	186 (55.5%)

**Table 3 ijerph-20-03969-t003:** Parental stress levels and socio-demographic variables.

	** *n(335)* **		
	**Frequency** **n (%)**	**Normal** **≤** **84%ile** **(n 139)**	**High-Clinically Significant ** **≥** **85%ile** **(n 196)**
Mother and caregiver characteristics			
Age of Mother			
≤35 y	200 (59.7%)	93 (46.5%)	107 (53.5%)
≥36 y	135 (40.3%)	46 (34.1%)	89 (65.9%)
Marital status			
Married	47 (14%)	19 (40.4%)	28 (59.6%)
Single	288 (86%)	120 (41.7%)	168 (58.3%)
Religion			
Christian	239 (71.3%)	101 (42.3%)	138 (57.7%)
Non-Christian	96 (28.7%)	38 (39.6%)	58 (60.4%)
Place of resident			
Urban	36 (10.8%)	19 (52.8%)	17 (47.2%)
Rural	299 (89.3%)	120 (40.1%)	179 (59.9%)
Level of education			
Matric & Tertiary	227 (67.8%)	96 (42.3%)	131 (57.7%)
No matric	108 (32.2%)	43 (39.8%)	65 (60.2%)
Employment			
Yes	50 (15.7%)	19 (38%)	31 (62%)
No	285 (84.3%)	120 (42.1%)	165 (57.9%)
Estimated income			
≥R5001	40 (11.9%)	14 (35%)	26 (65%)
R2001-R5000	10 (3%)	4 (40%)	6 (60%)
CDG	285 (85.1%)	111 (43.2%)	146 (56.8%)
Income source			
Salary	50 (15.5%)	18 (36.3%)	32 (60%)
Care dependency grant	285 (85.1%)	121 (42.5%)	164 (57.5%)
Availability of financial support			
Yes	304 (94.4%)	131 (43.1%)	173 (56.9%)
No	31 (5.6%)	8 (25.8%)	23 (74.2%)
Child related characteristics			
Age of child			
1–4 years	141 (42.1%)	58 (41.1%)	83 (58.9%)
5–8 years	120 (35.8%)	47 (39.2%)	73 (60.8%)
≥9 years	74 (22%)	34 (46%)	40 (54%)
Gender			
Girls	104 (30.9%)	53 (51%)	51 (49%)
Boys	231 (69.1%)	86 (37.2%)	145 (62.8%)
Local (KCD)			
Yes	236 (70.6%)	100 (42.4%)	136 (57.6%)
No	99 (29.4%)	39 (39.4%)	69 (60.6%)
Place of birth			
Hospital	285 (85.2%)	118 (41.4%)	167 (58.6%)
Clinic	38 (11.3%)	19 (50%)	19 (50%)
Home	12 (3.5%)	2 (16.7%)	10 (83.3%)
History of hospitalisation			
Yes	198 (59.4%)	91 (46%)	107 (54%)
No	137 (40.6%)	48 (35%)	89 (65%)
Multiple diagnosis			
Yes	200 (60.8%)	68 (34%)	132 (66%)
No	130 (39.2%)	70 (53.8%)	60 (46.2%)
Single diagnosis			
Cerebral Palsy	63 (18.7%)	30 (47.6%)	33 (52.4%)
Autism	31 (9.6%)	14 (45.2%)	17 (54.8%)
Communication	106 (31.8%)	30 (28.3%)	76 (71.7%)
Developmental delay	138 (41.4%)	45 (32.6%)	93 (67.4%)
Epilepsy	78 (23.2%)	35 (44.9%)	43 (55.1%)
Other diagnoses	108 (32.4%)	47 (43.5%)	61 (56.5%)
Health needs			
Rehabilitation	208 (62.2%)	86 (41.4%)	122 (58.6%)
Multiple Needs	215 (64.6%)	100 (46.5%)	115 (53.5%)
Age of initial Intervention			
≤2 years	247 (73.3%)	98 (39.7%)	149 (60.3%)
≥3 years	88 (26.7%)	41 (46.6%)	47 (53.4%)
Frequency of hospital visit			
Monthly	245 (73%)	86 (36.3%)	156 (63.7%)
Once in 2/6 months	90 (27%)	50 (55.6%)	40 (44.4%)
Tertiary level hospital visit			
Yes	196 (58.8%)	85 (43.4%)	111 (56.6%)
No	139 (41.2%)	54 (38.8%)	85 (61.2%)
Attending school			
Yes	108 (32.2%)	55 (49.1%)	55 (50.9%)
No	227 (67.8%)	86 (37.9%)	141 (62.1%)

**Table 4 ijerph-20-03969-t004:** Association between parental stress and the three subscales of the parental stress index.

	** *p* ** **-Value** **PSI-SF Subscales**				**Odds Ratio**	**95%** **CI**
	**PD**	**DC**	**P-CDI**	**PSI-Total**		
Mother and caregiver factors						
Age of Mother	* 0.00	0.1	0.3	* 0.02	1.8	1.09; 2.87
Place of resident	0.1	* 0.05	0.07	0.1	1.7	0.84; 3.45
Level of education	0.9	* 0.04	0.08	0.6	1.1	0.69; 1.89
Child related factors						
Gender of child	0.16	* 0.03	0.2	* 0.01	* 1.8	1.09; 2.79
History of hospitalisation	0.5	* 0.02	* 0.02	* 0.04	0.6	0.40; 0.99
Multiple diagnoses	0.1	* 0.00	* 0.01	* 0.00	* 2.3	1.44; 3.55
Communication difficulty	0.3	* 0.00	* 0.01	* 0.00	* 2.3	1.40; 3.77
Developmental delay	0.1	0.1	* 0.01	* 0.00	* 1.9	1.19; 2.93
Other disorders	0.4	* 0.03	* 0.05	0.5	0.9	0.55; 1.39
Multiple health needs	* 0.00	0.3	0.1	* 0.00	0.6	0.35; 0.89
Age of intervention	0.2	0.4	* 0.03	0.2	0.8	0.46; 1.23
Freq of a hospital visit	0.2	* 0.03	* 0.00	* 0.00	* 2.2	1.34; 3.58
Special hospital visit	0.09	0.07	0.2	* 0.00	0.8	0.53; 1.29
Attend school	* 0.03	0.3	* 0.00	* 0.04	* 1.2	0.98; 2.42

* statisticaly significant ≤ 0.05.

**Table 5 ijerph-20-03969-t005:** Multivariate logistic regression: Subscale level and total parental stress.

	Odds Ratio	Std. Err.	z	*p* > z	[95% CI]
Parent Distress(PD)					
Age of Mom	2.510198	0.6661333	3.47	0.001	1.49219; 4.222718
School	2.183299	0.5720926	2.98	0.003	1.306387; 3.648839
Difficult Child (DC)					
Multiple diagnosis	2.10799	0.5960893	2.64	0.008	1.211069; 3.669172
Communication	1.915711	0.6114521	2.04	0.042	1.024815; 3.581084
Freq hospital	1.788001	0.5194032	2.00	0.045	1.01181; 3.159632
Parental-Child Dysfunctional Interaction(P-CDI)					
Freq hospital	1.907087	0.5182993	2.38	0.018	1.119528; 3.248674
School	2.370937	0.659081	3.11	0.002	1.374992; 4.088273
Total Parental Stress Scores					
Age of Mother	2.308863	0.6352595	3.04	0.002	1.346477; 3.959108
Multiple diagnoses	2.01489	0.5679133	2.49	0.013	1.159667; 3.500818
School	1.978596	0.5639807	2.39	0.017	1.131697; 3.45927
Freq Hosp	1.93539	0.5690567	2.25	0.025	1.087658; 3.443853

## Data Availability

Data can be available if requested according to the data availability policy of Sefako Makgatho Health Sciences University.

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
