# Peer review of "Parental Stress in Raising a Child with Developmental Disabilities in a Rural Community in South Africa"

_ijerph, 2023, doi:10.3390/ijerph20053969_

Round 1

Reviewer 1 Report

Introduction:  The introduction contains information needed to understand the purpose, importance and meaning of the study, however it could be better organized. The issue of 'social/contextual' constructs is dispersed throughout.  A more focused story can be told in the Background

Methods: 

-Please provide more detail on the sample size calculation regarding the relationship between n and the PSI estimate.

-Briefly discuss the use/validity of the PSI in SA/Rural Africa

-Describe the demographic questionnaire-- since whether a child in school seems to be a salient factor, it would be helpful to understand age of school entry.

-Were there caregivers/parents who did not speak isiZulu? Any other exclusion/inclusion criteria?

-Describe how contextual/clinical factors were collected/determined. Were child characteristics by self-report from the caregiver?? 

Results

Can be streamlined into fewer tables.

The pie chart is not necessary.  Table 5 can be eliminated and 6&& can be combined.

Discussion/Conclusions

-There is no mention that a hospital-based sample was used. All caregivers had access to some medical support for their children. Another limitation--other than multiple diagnoses, no measure of severity of disability was captured.

Also, mention that there is no control group of families with no disability caregiving responsibilities 

Author Response

Dear Reviewer

Kindly please find attached response to the reviewer

Reviewer 2 Report

The purpose of this study was to examine parental stress in raising a child with developmental disabilities in a rural community in South Africa. Overall, the method is clear and appropriate, and the results are solid. I would like to point out the following as an attempt to further improve the manuscript contents:

1.    The sample of this study consists of mothers and caregivers. Could authors provide more information about the caregivers? For example, gender and relationships with the child.

2.    Some subtitles under the Study Methodology look repeated and redundant. For example, “Study design” looks similar to “Data collection” and “Data collection tool”. I would recommend combining these sessions.

3.    The limitation of the study needs to be extended. For example, due to the nature of cross-sectional design, the causal relationship of the effect cannot be concluded. In addition, the findings of this study (n=335) may not be generalized to all mothers and caregivers in this rural community in South Africa.

Author Response

Dear Reviewer

Kindly please find 

Regards

Round 2

Reviewer 1 Report

The additional information has improved the manuscript. Please also mention in the limitations that there is no comparison group, so the intro statement that parents of disabled children experience more stress than their counterparts whose children don't have a diagnosed disability cannot be explored.

Author Response

kindly, please find attached
